# Fire Control System Operation Status Assessment Based on Information Fusion: Case Study [note 1]

**DOI:** 10.3390/s19102222

**Published:** 2019-05-14

**Authors:** Yingshun Li, Aina Wang, Xiaojian Yi

**Affiliations:** 1Faculty of Electronic and Electrical Engineering, Dalian University of Technology, Dalian 116081, China; leeys@dlut.edu.cn (Y.L.); llldddyyy@mail.dult.edu.cn (A.W.); 2The School of Mechatronical Engineering, Beijing Institute of Technology, & Department of Overall Technology, China North Vehicle Research Institute & Academy of Mathematics and Systems Science, Chinese Academy of Sciences, Beijing 10071, China

**Keywords:** fire control system, status assessment, DS evidence theory, rough set theory, information fusion

## Abstract

In traditional fault diagnosis strategies, massive and disordered data cannot be utilized effectively. Furthermore, just a single parameter is used for fault diagnosis of a weapons fire control system, which might lead to uncertainty in the results. This paper proposes an information fusion method in which rough set theory (RST) is combined with an improved Dempster–Shafer (DS) evidence theory to identify various system operation states. First, the feature information of different faults is extracted from the original data, then this information is used as the evidence of the state for a diagnosis object. By introducing RST, the extracted fault information is reduced in terms of the number of attributes, and the basic probability value of the reduced fault information is obtained. Based on an analysis of conflicts in the existing DS evidence theory, an improved conflict evidence synthesis method is proposed, which combines the improved synthesis rule and the conflict evidence weight allocation methods. Then, an intelligent evaluation model for the fire control system operation state is established, which is based on the improved evidence theory and RST. The case of a power supply module in a fire control computer is analyzed. In this case, the state grade of the power supply module is evaluated by the proposed method, and the conclusion verifies the effectiveness of the proposed method in evaluating the operation state of a fire control system.

## 1. Introduction

Since the start of the 21st century, industrial manufacturing has gradually tended to become automated, large-scale, and systematic. In unmanned factories, the composition of industrial machinery and equipment is increasingly complex and functions increasingly well. Due to this, people are gradually paying more attention to the safety of equipment. The most commonly used method for traditional industrial equipment maintenance includes acquisition operation signal feature extraction and fault diagnosis. However, traditional fault diagnosis methods have been unable to adapt to the increasing tendency of industrial equipment to be: (1) modular; (2) generalized; (3) intelligent; and (4) modernized.

A fire control system is a kind of complex electronic system which is responsible for controlling the aiming and firing of weaponry. Such systems play an important role in modern warfare, and their development is becoming more and more intelligent and modernized. If the system fails, it will greatly reduce the combat capability of weaponry. A general-purpose fire control system has the characteristics of randomness, susceptibility, concurrency, and communication. Traditional fault diagnosis methods cannot meet the demand of fault diagnosis for a fire control system. For example, the state evaluation of the fire control computer and sensor subsystem is massive [1]. The state values from different levels reflect the running state of the fire control computer and sensor subsystem. However, due to the inaccurate measurement process and unified evaluation standards, the operational state evaluation of fire control computers and sensor subsystems has great ambiguity and uncertainty.

Since rough set theory (RST) contains a mechanism to deal with imprecise or uncertain problems, no prior information is needed to deal with uncertain information [2]. RST can objectively describe or deal with uncertain problems, avoid certain subjectivity, conduct attribute and value reduction for fault features, and use the reduced data for equipment fault prediction. Furthermore, Dempster–Shafer (DS) evidence theory has a strong ability to process uncertain information and thus one can employ the DS synthesis rule and diagnostic decision rule to perform decision analysis on the fusion reliability interval and obtain a final diagnosis result [3]. Additionally, there is evidence conflict in the evidence synthesis process of DS evidence theory, where the traditional approach’s synthesis method often abandons the conflict and proposes a multi-sensor data fusion method based on evidence reliability and entropy [4]. For the methods described above, there are still the following shortcomings: (1) the reasoning ability of RST is weak; (2) human factors often interfere with the evidence acquisition of DS evidence theory, which plays a crucial role in the decision-making results; and (3) the practice of abandoning the conflict part greatly reduces the reliability of the decision results. A multi-sensor data fusion method based on evidence reliability and entropy was proposed [5]; however, it is possible for the evidence combination to explode.

Based on the above research, this paper aims to tackle the problems in fault detection: the inability to synthesize conflicting evidence with the traditional synthesis rules and the combination explosion caused by excessive evidence dimensions. In this paper, a fire control system state evaluation method based on RST and improved evidence is proposed to solve the following problems: (1) the combination explosion; (2) the largely subjective nature of evidence acquisition; and (3) the inconsistency between the decision-making result and the actual result due to evidence conflicts.

In this paper, RST is first introduced to preprocess and discretize the collected fire control system data. In order to retain the core attributes, attribute reduction is carried out and the basic probability assignment is determined [6]. For the basic probability, based on the analysis of the existing DS rule improvement method and the conflict evidence weight allocation method, an improved conflict evidence synthesis method is proposed to effectively resolve the adverse impact of conflicting evidence on composite results. Thus, an intelligent operation state evaluation model of a fire control system based on the improved evidence theory is established. The data evaluation results based on the fire control computer power supply module show that the obtained evaluation results are consistent with prior knowledge. The results show that the model can more accurately evaluate the running state of the fire control system. Additionally, our approach can effectively reduce the adverse impact of uncertainty on the evaluation of the operation state of the fire control system. The power supply module of the fire control computer is taken as a case study to verify the effectiveness of the proposed method. The main results of our approach: (1) Obtain the set of simplest attributes by using RST; (2) the improvement of the DS evidence guarantees a low conflict evidence weight distribution, improves the degree of determining the weight distribution for high consistency and good conflict evidence, and ensures the consistency degree is high by assigning poor weight allocation to conflicting evidence; (3) The status evaluation results are consistent with the prior knowledge through the proposed information fusion method.

In Section 2, we describe the concept and principle of RST, including the attribute reduction algorithm based on the discernibility matrix; in Section 3, we describe the concept and principle of DS evidence, including the deficiencies of traditional evidence; in Section 4, we describe the improvement of the evidence; and in Section 5, we verify the validity of the proposed RST combined with the improvements to the traditional DS evidence theory method.

## 2. Rough Set Theory

The Polish scholar Z. Pawlak proposed RST in 1982. RST is an effective tool for dealing with incomplete information. It does not require any prior information except for the set being processed. By analyzing the relationship between the studied objects, the potential rules for the set are obtained [7].

### 2.1. The Basic concept of Rough Set Theory

#### 2.1.1. Information System and Decision Table

The information system is the primary structure in RST. The information system S=(U,A,V,f), where U=(x1,x2,…,xn) is the universe of discourse; A=C∪D is the set of attributes for all of the research objects [8]; where C=(c1,c2,…,cn) is the set of conditional attributes and D=(d1,d2,…,d5) is the set of decision attributes; *V* is the set of attribute values of the studied objects $V=\underset{a\in A}{\cup }V_{a}$, where Va is the value range for a∈A [9,10]; and *f*: U×A→V is the information function and is a single mapping, i.e., f(x,a)∈Va [11] specifies the value of the *x*th attribute for each object in U.

For S=(U,A,V,f), if the research object attribute set A consists of a conditional attribute C and a decision attribute *D*, that is, A=C∪D,C∩D=Φ, then the information system *S* is called the decision table [12].

#### 2.1.2. Indistinguishable Relation

Let S=(U,A,V,f) be an information system; B⊆U defines the indistinguishable relation *Ind*(*B*) about *B* as:(1)Ind(B)={(x,y)∈U×U:f(x,a)=f(y,a),∀a∈B}
The indistinguishable relation divides *U* into equivalence classes, where all objects in an equivalence class are indistinguishable from each other [13].

#### 2.1.3. Approximate Weight

The information system S=(U,A,V,f); *Ind*(*A*) is the equivalence relation over *U*; ∀X∈U, the approximate quality of set X with respect to attribute set *B*, is defined as:(2)$$\gamma B\left(X\right)=\frac{\underline{B}X}{U}$$where $\underline{B}X$ denotes the largest set of all subsets that must exist in the *X* set. The definition of approximate mass represents the probability that can be accurately divided under the attribute set *B*.

### 2.2. Attribute Reduction

Attribute reduction is performed to maintain the expressiveness of the information system and delete redundant attributes. Let R be an equivalence relation family. When *Ind*(*R*) = *Ind*(*R* − {r}), then r can be omitted from R, otherwise, r cannot be omitted from R. If r cannot be omitted for any m, then the family R is said to be independent. Obviously, when *R* is independent, and P⊆R, then *P* is independent [14]. The set of all the necessary relationships in *R* is called the core of *R*. The equivalence relation family is defined via: If the discretized values of each sample under any *n* attributes correspond to the same value, then the value of these *n* attributes in the decision table is equal and they are referred to as equivalent attributes.

#### 2.2.1. Discernibility Matrix

The discernibility matrix, also known as the discernible matrix or the distinct matrix, is a method of knowledge representation proposed by Skowron in 1991.

Let S=(U,A,V,f) be a decision table; for any x∈U, ∂(x) is the value of the object x on the attribute *C*$;\;d\left(x_{i}\right)$ is the decision attribute value corresponding to sample $x_{i}$. Then the discernibility matrix M(S)=[mij]n×n of S is [15,16]:(3)$$mij=\{\begin{array}{l} mij=0,d\left(xi\right)=d\left(xj\right)\\ mij=\left\{\partial |\partial \in C,\partial \left(xi\right)\neq \partial \left(xj\right)\right\},d\left(xi\right)\neq d\left(xj\right) \end{array}$$
It is obvious that the discernibility matrix is a symmetric matrix. Thus, in the calculation, it is only necessary to calculate its lower triangular matrix.

The above definition indicates that the differential elements of the *i*th row and *j*th column of M(S) are sets or values of conditional attributes. When the decision attribute values of $d\left(x_{i}\right)$ and $d\left(x_{j}\right)$ are different, then all of the conditional attributes that make xi and xj have different values constitute the difference element mij. The relation of these conditional attributes is the disjunction relation “∨”, meaning that any conditional attribute ∂ can distinguish xi from xj. If there is no conditional attribute that makes the xi and xj values different, that is, all of the conditional attribute values of xi and xj are the same, and the decision attribute values are different, then the difference element mij is an empty set.

Individuals with the same decision attribute value, whether or not the conditional attribute values are the same, cannot be distinguished. Therefore, when the decision attributes of individuals xi and xj are the same, the value of distinct elements is 0 instead of an empty set, indicating that there is no need to take it into consideration. Additionally, for the differential elements on the main diagonal of the discernibility matrix, for *i* = *j* and this xi = xj, the difference elements are also represented by 0 instead of the empty set φ.

#### 2.2.2. Discernibility Function

The discernibility function of the decision table S can be formally expressed as [17]:(4)fM(S)=∧{∨mij,1≤j≤i≤n,mij≠φ}
where ∨mij is the disjunction of the matrix term mij.

#### 2.2.3. Attribute Reduction Algorithm Based on the Discernibility Matrix

In order to avoid a large number of redundant attributes in the actual system which affect the speed of operation and accuracy of decision-making, this paper simplifies the decision table. An instance in the decision table can be considered as a rule, which may contain redundant attribute values, so the reduction of instance attribute values is the reduction of the decision rule [18]. The reduction of decision rules is the unnecessary condition of eliminating each rule separately. It is not the reduction of the attributes for the entire system, but the reduction of redundant attribute values for each decision rule. The formalization of the discernibility matrix is shown in Table 1.

Where $x_{i}$ is the element of sample, and the off diagonal values are the ∂ by applying the above Equation (3); the process for attribute reduction algorithm based on the discernibility matrix involves the following steps:Calculate the discernibility matrix *M(S)* for the decision table *S* according to the above Equation (3);Calculate the discernibility function *fM(S)* based on the discernibility matrix *M(S)*;Calculate the minimum disjunctive normal form of discernibility function *fM(S)*, where each disjunctive component is a reduction.

## 3. DS Evidence Theory

DS evidence theory is an uncertain reasoning method which can integrate the evidence provided by multiple evidence sources [19,20] and constantly reduces the hypothesis set based on the accumulation of evidence. It has strong decision-processing ability and is widely used in data fusion and target recognition [21].

### 3.1. Acquisition Method for the Basic Probability Value

Θ={θ1,θ2,…,θN} is a nonempty finite set; let Θ be the identification framework, if the functions $m:\;2^{\Theta }\to \left[0,1\right]$ satisfies the two conditions:

(1) m(⥂φ)=0 (where φ is the empty set), and (2) $\underset{A\subseteq \Theta }{\sum }m\left(A\right)=1$,

then, *m* is called the basic credibility distribution of the framework. Note, ∀A⊆Θ, *m(A)* is the basic credibility distribution of *A* and $2^{\Theta }$ is the power set of Θ. Essentially, m(A) is the probability of the occurrence of the statement *A*.

Assuming that *A* is any subset on the identification framework Θ, then the belief function $Bel:2^{\Theta }\to \left[0,1\right]$ satisfies [22,23]:(5)$$\begin{array}{l} Bel(\varphi )=m\left(\varphi \right)=0\\ Bel(\Theta )=\underset{B\subseteq \Theta }{\sum }m(B)=1\\ Bel(A1\cup A2\ldots \cup An)\geq \\ \underset{i}{\sum }Bel\left(Ai\right)-\underset{i}{\sum }Bel\left(Ai\cap Aj\right)+{\left(-1\right)}^{n}Bel\left(A1\cap A2\cap \ldots \cap An\right) \end{array}$$
If m(A)>0, then *A* is called the focal element of the belief function *Bel*. The relationship between the belief function *Bel*(A) and the basic credibility allocation function m is [24,25,26]:(6)$$Bel(A)=\sum_{B\subseteq A}m\left(B\right)$$
Based on Equation (6), it can be seen that the total trust degree of *A*—the sum of basic credibility—is expressed by *Bel*(A).

In the decision table S=(U,A,V,f), where *C* is the conditional attribute set and *D* is the decision attribute set [27,28], define U/D={Y1,Y2,…,Yn}. Thus, for ∀x∈U, the corresponding equation to calculate the basic credibility assignment is
(7)m(dix)=|[x]c∩Yj||[x]c|, i=1,2,…,n
where Yi={y|y∈U∧yD=di}; di∈VD.

### 3.2. Synthesis Method of Traditional Evidence Theory

DS evidence synthesis is based on the evidence decision coefficient. The general form of the evidence combination rule in DS evidence theory is [29]:(8)$$m\left(C\right)\{\begin{array}{l} m1⊕m2=\frac{\sum_{X\cap Y=Z}m1\left(X\right)m2\left(Y\right)}{1-K}\\ K=\underset{X\cap Y=\varphi }{\sum }m1\left(X\right)m2\left(Y\right)<1 \end{array}$$
where m1(X) and m2(Y) are the basic credibility functions of *X* and *Y*, respectively, and m(C) is the combined basic credibility function. The value of *K* indicates the degree to which the combined evidences conflict with each other. When K=0, it means that the two evidences are completely consistent (completely compatible); and when K=1, it means that the two evidences are in complete conflict. For 0<K<1, the two evidences are partially compatible.

### 3.3. Deficiencies of Evidence Theory

When applying evidence theory to practical situations, information with different degrees of conflict is often encountered. When dealing with different evidence bodies with low confidence and high conflict, the results are often not ideal and sometimes even contrary to common sense [30].

#### 3.3.1. Causes of Conflict

The main reasons why evidence theory produces phenomena that are inconsistent with common sense when dealing with contradictory evidence [31,32,33] are as follows. In order to maintain the normalization of the basic probability distribution function between evidences, the rule of evidence theory synthesis introduces the conflict factor *K*, which changes the basic probability distribution of the common focal element between the two evidences to 1/(1−K)  of the original probability distribution. However, conflicts are not generated by all focal elements together, and conflicts generated by common focal elements with large distribution functions are not necessarily large. Therefore, if the conflict information is completely abandoned without analysis, it will inevitably lead to the loss of important information.

#### 3.3.2. Classification of Conflict Problems

● Conventional Conflict Problem

High Conflict: Take the identification framework Θ={A,B,C} and the following evidence:
m1(A)=0.99; m1(B)=0.01; m1(C)=0;
m2(A)=0; m2(B)=0.01; m2(C)=0.99.
Using the traditional evidence theory synthesis rule, Equation (8), the synthesis results are shown in Table 2.

The conflict factor for this example is thus, K = 0.0099 + 0.9801 + 0.0099 = 0.09999, and m(B)=1.

The support degree for the two pieces of evidence is extremely small for *B*; however, the synthesis result suggests that *B* is fully supported. Obviously, this is inconsistent with the actual situation.

Total Conflict: assume the identification framework Θ={A,B} and the evidence as follows [34]:
m1(A)=0; m1(B)=1; 
m2(A)=1,m2(B)=0.
The conflict factor is K=1, the denominator of Equation (8) becomes 0, so the synthesis rule of evidence theory cannot be used [35].

● Less Robustness

High Conflict: Take the identification framework Θ={A,B,C} and the following evidence:
m1(A)=0.99; m1(B)=0.01; m1(C)=0;m2(A)=0; m2(B)=0.01; m2(C)=0.99.

After the synthesis of the two pieces of evidence, the result m(B)=0.01 was obtained, which was almost opposite to the synthesis result of m(B)=1. Therefore, when slight changes were found in the BPA of the focal element, the fusion result would be greatly different, which indicated that the synthesis rules of the evidence theory were very sensitive to the BPA of the focal element.

● One Vote Veto

Suppose the support degree of n pieces of evidence for A is as follows:
m1(A)=m2(A)=…mn(A)=0.99;
m3(A)=0.

The above n pieces of evidence are fused using the rule of evidence theory synthesis, and m(A) = 0. It can be seen from this example that, when one piece of evidence is inconsistent with or in complete conflict with multiple pieces of evidence, one negative vote will be obtained after the synthesis of the evidence, indicating that other evidence has no influence on the synthesis result.

## 4. Improvements to the Traditional Evidence Theory

Synthesis problems, in view of evidence theory, have been analyzed by many scholars, who have presented a series of effective solutions. Some of these approaches adopt a view that does not tally with the actual result for the synthesis rule. Holders of this viewpoint attempt to improve the evidence combination rules by using modified conflict information. Another view holds that the inaccuracy of the synthesis results derives from the evidence source rather than the synthesis rules of traditional evidence theory [36,37,38]. Additionally, there is a new view that the error of the result comes from an incomplete identification framework [39]. In this paper, evidence theory can be improved by modifying combination rules and evidence bodies [40].

### 4.1. Weight Distribution of Conflicting Evidence

At present, the weight distribution of conflicting evidence mainly adopts the conflict weight distribution method based on the distance between two pieces of evidence [41] or the similarity coefficient [42]. These commonly used methods usually only consider the mutual support degree between evidence and do not consider the role of evidence itself in decision making [43,44]. When the uncertainty degree of the evidence itself is high and the conflict caused by the evidence is low, the impact of the evidence on decision making is also low. Therefore, under the premise of good consistency between evidence and the decision-making, the less uncertain the evidence is, the more effective the decision will be. When the weight of conflict is allocated, the weight of the evidence should be higher, so as to make better decisions.

Let there be an identification frame Θ={θ1,θ2,…,θN}. The basic probability distribution of any event in the identification framework, which is presented in Equation (9), is converted into fuzzy membership μ:(9)$$\mu =\left[\begin{array}{l} \mu (\theta 1)\\ \mu (\theta 2)\\ \ldots \\ \mu (\theta n) \end{array}\right]=\left[\begin{array}{l} Bel(\theta 1)\\ Bel(\theta 2)\\ \ldots \\ Bel(\theta n) \end{array}\right]=\left[\begin{array}{l} \sum_{\alpha 1\subseteq \theta 1}m(\alpha 1)\\ \sum_{\alpha 2\subseteq \theta 2}m(\alpha 2)\\ \ldots \\ \sum_{\alpha n\subseteq \theta n}m(\alpha n) \end{array}\right]$$
Then, the distance between any two evidences is defined as:(10)$$d\left(m1,m2\right)=1-\;\frac{\sum_{i=1}^{n}\left(\mu^{\left(1\right)}\left(\theta i\right)\cap \mu^{\left(2\right)}\left(\theta i\right)\right)}{\sum_{i=1}^{n}\left(\mu^{\left(1\right)}\left(\theta i\right)\cup \mu^{\left(2\right)}\left(\theta i\right)\right)}$$
where ∩ represents intersection (smaller value) and ∪ represents union set (larger value). In order to avoid the scenario that the denominator of the fraction in the equation is 0, it is required to identify focal elements in the framework. The gap between two evidence bodies is represented by the evidence distance, which is another way of expressing evidence conflict. When the distance between evidences is 0, it means that there is no conflict between the two pieces of evidence, and they are completely consistent. That is, the degree of consistency between the two evidences is 1. When the distance between the evidences is 1, it means that there is complete conflict between the two evidences and there is no similarity, that is, the degree of consistency between the two evidences is 0.

Thus, the similarity coefficient of the two pieces of evidence is
(11)$$s\left(m1,m2\right)=1-d\left(m1,m2\right)=\frac{\sum_{i=1}^{n}\left(\mu^{\left(1\right)}\left(\theta i\right)\cap \mu^{\left(2\right)}\left(\theta i\right)\right)}{\sum_{i=1}^{n}\left(\mu^{\left(1\right)}\left(\theta i\right)\cup \mu^{\left(2\right)}\left(\theta i\right)\right)}$$

Assuming that the number of evidences obtained by the system is *m*, the evidence similarity number can be calculated by using the above formula, and the similarity matrix is thus:(12)$$s=\left[\begin{array}{llll} 1 & d12 & \ldots & d1n\\ d21 & 1 & \ldots & d2n\\ \ldots & \ldots & \ldots & \ldots \\ dn1 & dn2 & \ldots & 1 \end{array}\right]$$

The degree of consistency between two evidences reflects the degree of mutual support between the two pieces of evidence [45,46]; therefore, the support degree of each piece of evidence for Ej can be obtained by adding the rows or columns of the similarity matrix as follows [47]:(13)$$Sup(mj)=\overset{n}{\underset{j=1}{\sum }}dij\left(i,j=1,2,\ldots ,n\right)$$

Obviously, the greater the degree to which a piece of evidence is supported by other evidence, the higher its credibility will be, and vice versa. Therefore, inter-evidence support is often used to represent the credibility of a single piece of evidence. Equation (14) is normalized to calculate the credibility of the evidence, as shown in
(14)$$Crd(mi)=\frac{Sup\left(mi\right)}{\sum_{i}^{n}Sup\left(mi\right)}\left(i,j=1,2,\ldots ,n\right)$$

In recent decades, various uncertainty measures have been proposed, e.g., Yager’s dissonance measure, Shannon entropy, and others derived from the Boltzmann–Gibbs (BG) entropy in thermodynamics and statistical mechanics, which have been used as an indicator to measure uncertainty associated with a probability density function (PDF). In this paper, a new entropy, called the Deng entropy, is proposed to handle the uncertain measure of Basic Probability Assignment (BPA). Deng entropy can be seen as a generalized Shannon entropy [48]. When the BPA is degenerate with the probability distribution, the Deng entropy is degenerate with the Shannon entropy. Benefiting from the above research, the uncertainty measure of probability has a widely accepted solution [49]; assuming that the recognition framework contains *n* pieces of evidence, which yields the corresponding basic probability assignment, the Deng entropy of the *i*th piece of evidence is defined as follows: (15)$$Ed\left(mi\right)=-\overset{N}{\underset{i=1}{\sum }}m\left(\theta i\right)\log2\frac{m\left(\theta i\right)}{2^{\left|\theta i\right|}-1}$$
where *m* is a mass function defined on the frame of discernment Θ, θi is a focal element of *m*, and |θi| is the cardinality of θi. As shown in the above definition, Deng entropy is formally similar to the classical Shannon entropy, however, the belief for each focal element θi is divided by a term $2^{\left|\theta i\right|}-1$, which represents the potential number of states in θi (of course, the empty set is not included). Through a simple transformation, it is found that Deng entropy is actually a type of composite measure, as follows:(16)$$Ed\left(mi\right)=-\sum_{i=1}^{N}m\left(\theta i\right)\log2(2^{\left|\theta i\right|}-1)-\sum_{i=1}^{N}m\left(\theta i\right)\log2m\left(A\right)$$
where the term $\sum_{i=1}^{N}m\left(\theta i\right)\log2(2^{\left|\theta i\right|}-1)$ is interpreted as a measure of the total non-specificity in the mass function *m*, and the term $-\sum_{i=1}^{N}m\left(\theta i\right)\log2m\left(A\right)$ is a measure of the discord of the mass function among various focal elements.

Deng entropy can measure the uncertainty degree of some information. The higher the Deng entropy, the higher the degree of uncertainty. In a decision-making system, the more uncertain the information is (high Deng entropy), the lower its influence on decision making should be. Conversely, the more certain the information is (low Deng entropy), the higher the impact on decision making should be. Evidence consistency is used to reflect this idea in the overall decision-making process, as well as the degree of conflict with other evidence. The higher the consistency of evidence, the lower the degree of conflict with other evidence, and the more likely it will be to yield correct decisions. In the decision system, the average basic probability assignment of each subset (decision attribute) in the identification framework can reflect the correct trend of decision making in the evidence set. If there is a large distance between the basic probability assignment [50] of a subset in the identification framework and the average basic probability assignment, the evidence consistency is low. Conversely, if there is a small distance between the two, the evidence consistency is higher.

For an identification framework Θ={θ1,θ2,…,θN} with n pieces of evidence, the recognition framework is E1,E2,…,EN, and m1,m2,…,mn is the corresponding basic probability assignment. Thus, the evidence consistency is defined as [51]:(17)$$Con\left(mi\right)=1-\sqrt{\sum_{j=1}^{N}{\left(mi\left(\theta j\right)-\bar{m\left(\theta j\right)}\right)}^{2}},\;\bar{m\left(\theta j\right)}=\frac{\sum_{i=1}^{n}mi\left(\theta j\right)}{n}$$
where $\bar{m\left(\theta j\right)}$ represents the average basic probability assignment of each subset within the recognition framework.

Evidence validity shows the influence of evidence on making correct decisions, which can be comprehensively measured by the consistency of evidence and the certainty of evidence. When the degree and consistency of evidence are high, it indicates that the evidence is of great help to make correct decisions and has high effectiveness. When the degree of evidence confirmation is high and the consistency is low, although the evidence has a high role in decision making, its effectiveness should be low, so as to avoid the adverse impact of increasing evidence on making correct decisions. The evidence validity can be defined as:(18)$$Eff\left(mi\right)=Con\left(mi\right)\times e^{-Edi}$$

Since evidence validity reflects the influence of the evidence itself on correct decision making, and the credibility is based on the similarity between one piece of evidence other evidence, this paper redefines the weight distribution of conflicting evidence by using both evidence validity and evidence credibility as follows:(19)$$\omega i=\frac{Eff\left(mi\right)\times Crd\left(mi\right)}{\sum_{i=1}^{n}\left[Eff\left(mi\right)\times Crd\left(mi\right)\right]}$$

The weight distribution method of conflicting evidence increases the weight of conflicting evidence that can help make correct decisions and reduces the weight of conflicting evidence that may make wrong decisions, so as to achieve better convergence results.

### 4.2. Steps to Improve the Method

According to the new conflict evidence weights, an improved conflict evidence synthesis method can be obtained. The specific steps of this method are as follows:

Step 1: For the *n* pieces of evidence within the recognition framework, the similarity coefficients between every two pieces of evidence are calculated, and the similarity matrix is obtained.

Step 2: Calculate the support degree of the *i*th piece of evidence according to the similarity matrix. After normalization of the support degree, obtain the credibility for each evidence.

Step 3: Calculate the information entropy and consistency of each evidence and obtain the evidence validity according to Equation (18).

Step 4: Considering the influence of evidence credibility and evidence validity on conflicting evidence, the weight of the piece of evidence is calculated according to Equation (19).

Step 5: According to this new weight distribution for the conflict evidence, redefine the conflict distribution function *f*(A) to obtain a new conformity rule, via:(20)$$f\left(A\right)=K\times q\left(A\right)=K\times \overset{n}{\underset{i=1}{\sum }}(\omega i\times mi\left(Aj\right))$$
(21)$$m(A)=\{\begin{array}{l} m\left(A\right)=0,A=\Phi \\ m\left(A\right)=\underset{\cap Aj=A}{\sum }\underset{1\leq i\leq n}{\prod }mi\left(Aj\right)+f\left(A\right),A\neq \Phi \end{array}$$
where $K=\sum_{\cap Aj=\Phi }\prod_{1\leq i\leq n}mi\left(Aj\right)$.

### 4.3. The Performance of Our Proposed Synthesis Method Compared to Other Methods

In order to verify the effectiveness of the new conflict evidence synthesis rule designed in this paper, a comparative analysis was conducted with the basic probability assignment given by the high conflict evidence in Table 3, as an example. As the conflict factor K=1, the classical DS synthesis method cannot be used. We compare our proposed approach to several other methods briefly introduced below.

Yager’s formula [52] removes the normalization factor and assigns the conflict information generated by the evidence to the identification framework Θ. The rules of composition are as follows:(22)$$m\left(A\right)=\{\begin{array}{l} \underset{Aj\cap Bj\cap CK\cap \ldots =A}{\sum }m1\left(Aj\right)m2\left(Bj\right)ml\left(CK\right)\ldots ,A\neq \Theta \\ \underset{Aj\cap Bj\cap CK\cap \ldots =A}{\sum }m1\left(Aj\right)m2\left(Bj\right)ml\left(CK\right)\ldots ,+K,A=\Theta \end{array}$$
(23)m(ϕ)≠0
where $K=\sum_{Aj\cap Bj\cap CK\cap \ldots =\varphi }m1\left(Aj\right)m2\left(Bj\right)ml\left(CK\right)\ldots$. This method has the problem of “one vote no”, and the result is not ideal when multiple evidence bodies are synthesized.

By defining the concept of evidence credibility, Sun Quan et al. [53] set the evidence conflict parameter K, average support degree $\bar{m\left(\theta j\right)}$, and credibility parameter ε. They further used the credibility to allocate the parts of the conflict information assigned to unknown items. Suppose that the evidence E1,E2,…, En corresponding to m1,m2,…,mn is as follows [54]:(24)$$m\left(\theta \right)=\{\begin{array}{l} p\left(\theta \right)+K\times \varepsilon \times \bar{m\left(\theta j\right)},\theta \neq \Theta \\ p\left(\theta \right)+K\times \varepsilon \times \bar{m\left(\theta j\right)}+K\left(1-\varepsilon \right),\theta =\Theta \end{array}$$
(25)m(ϕ)=0
With $p\left(\theta \right)=\sum_{\theta i\in Ei,\underset{i=1}{\cap }\theta i=\theta }m1\left(\theta 1\right)m2\left(\theta 2\right)\ldots mn\left(\theta n\right)$ and $\bar{m\left(\theta j\right)}=\frac{1}{n}\sum_{i=1}^{n}mi\left(\theta \right)$.

The smaller the conflict parameter *K*, the less the conflict between pieces of evidence will be. The fusion result is mainly determined by p(θ). When the evidence conflict is larger or there is complete conflict, *K* is closer to 1, and the results of the approach average the credibility and the support degree Furthermore, the conflict between the distribution of the information is related to the evidence on the degree of the average support θ. Thus, the probability of the unknowns dominates the distribution results against our judgment and analysis, and this method is not reasonable to effectively deal with conflict problems.

In Li Bicheng’s opinion [55], conflicting evidence can all be used, and an average distribution of evidence is defined. However, the average idea proposed by this method is not consistent with reality.

In the literature [56], conflicts caused by evidence are allocated to conflicting focal points according to the BPA proportional relation of conflicting focal points [56]. The commutativity of the rules of evidence theory synthesis cannot be satisfied. When the order of synthesis is different, the results of synthesis are different, which limits the application of the rules of evidence theory synthesis.

In this paper, the above synthesis method is adopted. To compare and illustrate the effectiveness of the evidence synthesis method and the rationality of the fusion result, we take four samples m1, m2, m3, m4 and the basic probability assignment of high conflict evidence as an example for comparative analysis, as shown in Table 3. The results are shown in Table 4. *m*(A), *m*(B), *m*(C) respectively represents the basic probability distribution function of A, B, C. Where m1,2,3,4(A) is the support for target A after synthesis of evidence m1, m2, m3, m4, m(Θ) is the fundamental probability of the unknown.

As can be seen from the results in Table 4, both Yager’s method and the method from Reference [54] assign the conflict probability to unknown terms, and the fusion result cannot make a correct decision. The methods in References [55] and [56] modify the composition rules, assign the conflict to the basic probability of the conflict focal element, improve the rationality of the composition results, and can thus help one make a correct decision. By introducing the weight of evidence for effectiveness, the method in this paper improves on the other methods. Our approach guarantees a low conflict evidence weight distribution, improves the degree of determining the weight distribution for high consistency and good conflict evidence, and ensures the consistency degree is high by assigning poor weight allocation to conflicting evidence. Thus, our approach has better convergence effects, is more accurate, and can be used to make correct decisions.

## 5. Case Analysis

### 5.1. Establishment of a Diagnosis Model

For the state assessment and fault diagnosis of a weapon fire control system, rough set and DS evidence theory are introduced. The rough set restores the information system formed by the original data and extracts the classification rules so that the evidence theory can obtain the basic credibility. Then by allocating and using the synthesis rules, the final decision result is determined. The basic structure of the data fusion method based on rough set and evidence theory is shown in Figure 1.

### 5.2. Diagnostic Examples

Take the power supply module for the core subsystem of a certain type of tank fire control, fire control computer, and sensor subsystem as an example. The power supply module primarily shows a 26 V main signal that is powered by an external 5 A fuse and is used for the fire control computer through a filter. The 26 V power supply(01) on the power board and the 26 V power supply(02) to the sighting system generate a ±15 V power signal on the power board. The application of a letter-following fusion method is used to construct and analyze the power supply module, and the validity and practicability of the algorithm are verified.

#### 5.2.1. Modeling

This article has five training sample objects, 13 conditional attributes ci(i=1,2,…,13), and one decision attribute D. Table 5 shows the set of state evaluations for the decision attribute, D={d1,d2,d3,d4,d5}, where the normal state code is 1, the power supply state of ±15 V is denoted 2, the failure state of the 26 V power supply 01 is denoted 3, the failure state of the 26 V power supply 02 is denoted 4, and the failure of the main 26 V power supply is denoted 5. Where 26 V (01) power supplies the power panel of fire control computer; and 26 V (02) power supplies the laser power counter, sight control box, and sight mirror body.

The set of conditional attribute codes is C={c1,c2,…,c13} and is shown in Table 6.

Where the corresponding pin is the aviation plug port number, where Power 15 V (a) and Power 15 V (b) are used to power supply ADA-5a board, Power-15V (a) and Power-15V (b) are used to power supply ADA-5b board.

Set the research sample object as U={x1,x2,…,x20}. The original data is shown in Table 7. The raw voltages data is measured by connecting the upper computer with the power module through the test cable.

The conditions of the discretization according to the experience of experts are shown in Table 8.

Discretization of data is completed; the process of discretization involves the following equation:$$Discrete\;value\left(v\right)\{\begin{array}{l} 0,raw\;voltage\;is\;less\;than\;the\;minimum\\ 1,raw\;voltage\;is\;less\;than\;the\;maximum\;and\;more\;than\;the\;minimum\\ 2,raw\;voltage\;is\;more\;than\;the\;maximum \end{array}$$
and the generated discretized decision table can be used to handle attribute reduction using rough set theory; the results are shown in Table 9.

#### 5.2.2. Redundant Attribute Reduction

From the original data table, we find all the equivalent attributes in the decision table and perform reduction based on the defined equivalent attributes. The discovered equivalent attributes are as follows:
h1={c1,c2,c3,c9}, h2={c4,c10,c11,c12,c13}, h3={c5}, h4={c6,c7}, and h5={c8}.
After removing the redundant attributes, the conditional attribute set becomes R={c1, c4, c5, c6, c8}.

According to the approximate mass equation by applying the above Equation (2), we next find the approximate quality of the conditional attributes in C:
$$\gamma c1=0,\;\gamma c4=0,\;\gamma c5=\frac{1}{10},\;\gamma c6=\frac{1}{20},\;\gamma c8=\frac{3}{20}$$

After further attribute reduction, the reduction set is R={c5, c6, c8} and the 13 conditional attributes are reduced to only three remaining important attributes. This fully demonstrates the reduction ability of rough set theory, and thus, the processing efficiency of the subsequent DS evidence theory is improved [57]. Attribute reduction should be maintained in principle. Before and after the reduction, the integrity of the content of the decision table must be ensured.

#### 5.2.3. Allocation of the Fundamental Credibility

Based on Table 10, by applying the above Equation (7), the basic credibility for all of the evidence is assigned, as shown in Table 11. Table 10 is also the beginning of the combination of DS evidence theory and rough set theory.

According to Equation (21), for the conflict factor, a value of K=0.9999 is obtained. The synthesis results are summarized in Table 12 and Table 13, which adopt the classical DS theoretical synthesis method.

Thus, from Table 13, we find m(d1)=0.1183, m(d2)=0.1773, m(d3)=0.3188, m(d4)=0.1773, m(d5)=0.2083. From these results, the diagnosis result is d3; however, the actual operation level is d5; the actual operation level is not obtained and the diagnosis result is invalid. Therefore, the classical DS synthesis method is unable to evaluate the reliability of the power module under the conditions of high conflict. Additionally, Yager’s method and the method from Reference [52] will assign the probability of conflict to unknown terms, and the synthetic results cannot accurately give the evaluation results of the running state.

Thus, the improved conflict evidence synthesis method presented in this paper is adopted to synthesize such high conflict evidence. The synthesis steps are as follows:

Step 1: Calculate the similarity matrix by applying the above Equation (12):$$S=\left[\begin{array}{lll} 1 & 0.8347 & 0.9465\\ 0.8347 & 1 & 0.9321\\ 0.9465 & 0.9321 & 1 \end{array}\right]$$

Step 2: Calculate the support degree between pieces of evidence by applying the above Equation (13):Sup(m1)=2.7812,Sup(m2)=2.7668,Sup(m3)=3.8251.

Step 3: Calculate the credibility of the evidence by applying the above Equation (14):$$C\mathrm{rd}(m1)=\frac{Sup\left(m1\right)}{\sum_{i=1}^{3}Sup\left(mi\right)}=0.2967,\;C\mathrm{rd}(m2)=\frac{Sup\left(m2\right)}{\sum_{i=1}^{3}Sup\left(mi\right)}=0.2952,\;C\mathrm{rd}(m3)=\frac{Sup\left(m3\right)}{\sum_{i=1}^{3}Sup\left(mi\right)}=0.4081.$$

Step 4: Calculate the Deng entropy of the evidence by applying the above Equation (15):Ed1=1.5903,Ed2=1.5992,Ed3=1.2882;
calculate the consistency of evidence by applying the above Equation (17):Con(m1)=0.8794,Con(m2)=0.9581,Con(m3)=0.8446;
and calculate the validity of evidence by applying the above Equation (18):Eff(m1)=0.1793,Eff(m2)=0.1936,Eff(m3)=0.2329.

Step 5: Calculate the weights of the conflict evidence by applying the above Equation (19):ω1=0.2921,ω2=0.3586,ω3=0.4314;
and calculate q(θ) by applying the above Equation (20):q(θ1)=0.1688,q(θ2)=0.2009,q(θ3)=0.2023,q(θ4)=0.2009,q(θ5)=0.2973.

Thus, according to Equation (20), the following result is obtained: m(θ1)=0.1688, m(θ2)=0.2008, m(θ3)=0.2023, m(θ4)=0.2008, m(θ5)=0.2973.

Since m(θ5)=0.2973 is the largest value, the corresponding operation level is d5. That is, the 26 V main power failure hidden state is the predicted operating state, and the diagnostic result of test sample is consistent with the actual processing result. The method designed in this paper is used to evaluate the operation state of the monitoring data (fault unknown) of the power supply module of a certain type of fire control computer.

## 6. Conclusions

This article first introduced rough set theory and the related concepts of DS evidence theory. Then, we studied how the combination of rough set theory and DS evidence theory could play to the advantages of the two algorithms and avoid their disadvantages. An analysis of the classical theory of evidence conflict and evidence synthesis rules was performed. On the basis of an evidence weight allocation method, this paper proposed an evidence similarity calculation method using the fuzzy membership degree of reliability and the validity of the evidence. Furthermore, an improved conflict evidence synthesis method was proposed and studied with a comparative analysis. Finally, the improved method was applied to the intelligent evaluation model of a power supply module of a fire control computer. An example shows that this model can solve the paradoxical problem caused by the conflict of evidence theory synthesis rules to some extent and can accurately evaluate the running state.

There is a large amount of running state data for a fire control computer. In the future, we suggest establishing an overall running state evaluation system according to the classification of incoming and outgoing signals. By gradually improving the study of state evaluation, one can meet the required standards for state evaluation of a fire control system and provide a favorable basis for the realization of situational maintenance.

## Figures and Tables

**Figure 1 sensors-19-02222-f001:**
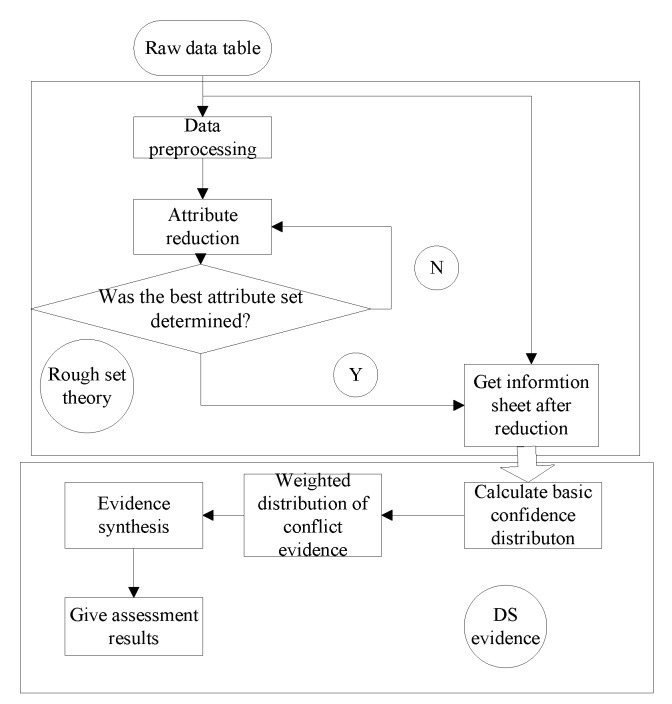
Structure of data fusion based on rough set and evidence theory.

**Table 1 sensors-19-02222-t001:** Schematic diagram of the discernibility matrix.

	x1	x2	x3	…	xn
x1	0				
x2	mij	0			
x3	mij	mij	0		
…	mij	mij	mij	0	
xn	mij	mij	mij	mij	0

**Table 2 sensors-19-02222-t002:** Synthesis results of the traditional evidence theory from the example above.

	m1(A)	m1(B)	m1(C)
m2(A)	0	0	0
m2(B)	0.0099	0.01	0
m2(C)	0.9801	0.0099	0

**Table 3 sensors-19-02222-t003:** The basic probability assignment for high conflict evidence.

Evidence	m1	m2	m3	m4
*m*(A)	1	0.46	0	0.9
*m*(B)	0	0.49	1	0.1
*m*(C)	0	0.05	0	0

**Table 4 sensors-19-02222-t004:** Comparison of synthesis results for the example in Table 4.

Method	m1,2,3,4(A)	m1,2,3,4(B)	m1,2,3,4(C)	m(Θ)
Yager’s method [52]	0	0	0	1
Method in Reference [53]	0	0	0	1
Method in Reference [55]	0.59	0.3975	0.0125	0
Method in Reference [56]	0.6851	0.2996	0.0153	0
Method in this paper	0.738	0.246	0.016	0

**Table 5 sensors-19-02222-t005:** Decision attribute codes for various states (D).

No.	State Name	Decision Attribute Coded Value
d1	Normal state	1
d2	±15 V power failure hidden status	2
d3	26 V (01) power failure hidden status	3
d4	26 V (02) power failure hidden status	4
d5	26 V main power failure hidden state	5

**Table 6 sensors-19-02222-t006:** Conditional attribute codes (C).

No.	Parameter Name and Unit	Corresponding Pin
c1	Power 15V (a)	XS2–15
c2	Power 15V (b)	XS2–20
c3	Power-15V (a)	XS2–16
c4	Power-15V (b)	XS2–21
c5	Power26V (01)	XS2–25
c6	Power26V (02a)	XS3–1
c7	Power26V (02b)	XS3–2
c8	Power26V (02c)	XS4–18
c9	Power26V (02d)	XS4–38
c10	Main Power26V (a)	XS2–34
c11	Main Power26V (b)	XS2–35
c12	Main Power26V (c)	XS2–36
c13	Main Power26V (d)	XS2–37

**Table 7 sensors-19-02222-t007:** Raw voltages measured in various states of the fire control system.

U c(v)	c1	c2	c3	c4	c5	c6	c7	c8	c9	c10	c11	c12	c13	D
x1	−16.95	−16.95	−7.8	−14.92	23.14	21.43	21.53	24.31	0.02	24.32	24.32	24.32	24.28	1
x2	−16.95	−16.95	−7.75	−14.92	23.46	21.52	21.48	24.75	−0.01	24.71	24.67	24.65	24.61	1
x3	−16.95	−16.95	−7.76	−14.92	23.39	23.55	23.46	−0.01	0.00	24.65	24.67	24.64	24.56	1
x4	−14.93	−15.48	−7.69	−17.21	25.12	21.55	22.05	26.02	0.00	25.47	25.74	25.51	28.30	1
x5	−16.49	−15.54	−7.76	−15.17	25.38	21.85	20.74	23.89	0.01	25.46	24.46	26.86	26.25	2
x6	−16.68	−13.99	−7.53	−13.98	24.24	21.68	23.30	28.12	0.02	26.32	28.07	28.25	27.18	2
x7	−16.68	−16.08	−7.77	−14.61	25.81	22.88	20.71	25.43	−0.01	26.24	27.36	27.61	26.39	2
x8	−17.01	−15.70	−7.52	−14.61	24.50	22.74	23.16	24.21	−0.02	25.35	27.98	25.91	23.53	2
x9	−16.30	−13.64	−7.68	−16.71	24.35	21.77	22.06	27.74	0.01	25.18	24.88	27.33	25.72	3
x10	−13.80	−17.33	−7.75	−16.28	25.27	22.62	22.37	25.20	0.02	24.68	26.90	25.56	26.76	3
x11	−17.29	−13.60	−7.70	−13.92	23.38	21.77	23.06	23.56	0.00	25.08	26.85	25.91	26.11	3
x12	−14.52	−17.33	−7.72	−14.75	22.56	22.62	21.79	24.52	0.01	27.63	24.03	27.35	25.33	3
x13	−15.96	−13.60	−7.71	−15.32	23.41	23.42	23.46	23.89	0.00	24.00	25.51	25.56	26.48	4
x14	−16.87	−16.74	−7.76	−14.42	24.91	22.90	20.88	24.54	−0.01	25.67	24.83	28.45	27.93	4
x15	−14.95	−14.83	−7.71	−14.55	24.23	22.65	20.58	24.89	−0.01	25.68	27.12	28.53	28.25	4
x16	−15.89	−15.15	−7.50	−16.13	25.54	20.74	20.57	25.23	0.00	26.90	24.87	27.89	26.87	4
x17	−16.72	−14.79	−7.67	−17.31	23.62	21.64	22.31	24.75	−0.01	23.49	28.06	25.42	24.47	5
x18	−15.75	−16.05	−7.60	−17.28	23.97	22.29	22.00	25.52	0.01	27.14	27.69	25.76	26.80	5
x19	−17.3	−15.01	−73.7	−14.33	25.27	21.87	22.54	26.12	0.01	27.11	25.52	24.68	23.77	5
x20	−16.38	−14.25	−7.67	−13.90	23.31	20.56	21.09	25.53	0.00	25.77	25.98	24.47	27.71	5

**Table 8 sensors-19-02222-t008:** Discretization standards.

Conditional Attributes	Discrete Value (V)		
	0	1	2
c1	<13.5	13.5 to 16.5	>16.5
c2	<13.5	13.5 to 16.5	>16.5
c3	<−17.5	−17.5 to –13.5	>−13.5
c4	<−17.5	−17.5 to–13.5	>−13.5
c5	<23.4	23.4 to 28.6	>28.6
c6	<23.4	23.4 to 28.6	>28.6
c7	<23.4	23.4 to 28.6	>28.6
c8	<23.4	23.4 to 28.6	>28.6
c9	<23.4	23.4 to 28.6	>28.6
c10	<23.4	23.4 to 28.6	>28.6
c11	<23.4	23.4 to 28.6	>28.6
c12	<23.4	23.4 to 28.6	>28.6
c13	<23.4	23.4 to 28.6	>28.6

**Table 9 sensors-19-02222-t009:** Discretized decision table.

U	c1	c2	c3	c4	c5	c6	c7	c8	c9	c10	c11	c12	c13	D
x1	0	0	0	1	0	0	0	1	0	1	1	1	1	1
x2	0	0	0	1	1	0	0	1	0	1	1	1	1	1
x3	0	0	0	1	0	1	1	0	0	1	1	1	1	1
x4	0	0	0	1	1	0	0	1	0	1	1	1	1	1
x5	0	0	0	1	1	2	2	2	0	1	1	1	1	2
x6	0	0	0	1	1	0	0	1	0	1	1	1	1	2
x7	0	0	0	1	1	0	0	1	0	1	1	1	1	2
x8	0	0	0	1	2	0	0	1	0	1	1	1	1	2
x9	0	0	0	1	1	0	0	2	0	1	1	1	1	3
x10	0	0	0	1	1	0	0	1	0	1	1	1	1	3
x11	0	0	0	1	1	0	0	1	0	1	1	1	1	3
x12	0	0	0	1	0	0	0	1	0	1	1	1	1	3
x13	0	0	0	1	1	1	1	1	0	1	1	1	1	4
x14	0	0	0	1	2	0	0	1	0	1	1	1	1	4
x15	0	0	0	1	1	0	0	2	0	1	1	1	1	4
x16	0	0	0	1	1	0	0	1	0	1	1	1	1	4
x17	0	0	0	1	1	0	0	1	0	1	1	1	1	5
x18	0	0	0	1	0	0	0	1	0	1	1	1	1	5
x19	0	0	0	1	1	0	0	1	0	1	1	1	1	5
x20	0	0	0	1	0	0	0	1	0	1	1	1	1	5

**Table 10 sensors-19-02222-t010:** Reduced decision table.

U	Conditional Attribute Discrete Value				U	Conditional Attribute Discrete Value			
	c5	c6	c8	D		c5	c6	c8	D
x1	0	1	1	1	x11	1	1	1	3
x2	1	1	1	1	x12	0	1	1	3
x3	0	0	0	1	x13	1	0	1	4
x4	1	1	1	1	x14	2	1	1	4
x5	1	2	2	2	x15	1	1	2	4
x6	1	1	1	2	x16	1	1	1	4
x7	1	1	1	2	x17	1	1	1	5
x8	2	1	1	2	x18	0	1	1	5
x9	1	1	2	3	x19	1	1	1	5
x10	1	1	1	3	x20	0	1	1	5

**Table 11 sensors-19-02222-t011:** Basic trust allocation.

Conditional Attribute	Basic Probability	Conditional Attribute Discrete Value	Decision Attribute Value(D)				
			1	2	3	4	5
c5	m1	0	$\frac{2}{5}$	0	$\frac{1}{5}$	0	$\frac{2}{5}$
		1	$\frac{2}{13}$	$\frac{3}{13}$	$\frac{3}{13}$	$\frac{3}{13}$	$\frac{2}{13}$
		2	0	$\frac{1}{2}$	0	$\frac{1}{2}$	0
c6	m2	0	$\frac{1}{2}$	0	0	$\frac{1}{2}$	0
		1	$\frac{3}{17}$	$\frac{3}{17}$	$\frac{4}{17}$	$\frac{3}{17}$	$\frac{4}{17}$
		2	0	1	0	0	0
c8	m3	0	1	0	0	0	0
		1	$\frac{3}{16}$	$\frac{3}{16}$	$\frac{3}{16}$	$\frac{3}{16}$	$\frac{4}{16}$
		2	0	$\frac{1}{3}$	$\frac{1}{3}$	$\frac{1}{3}$	0

**Table 12 sensors-19-02222-t012:** Results of the fusion of m1 and m2 into m1,2.

Evidence	1	2	3	4	5
*m*1,2	0.12	0.20	0.30	0.20	0.18
*m*3	0.1875	0.1875	0.1875	0.1875	0.25

**Table 13 sensors-19-02222-t013:** Results of the fusion of m1, m2, and m3 into m1,2,3.

Evidence	1	2	3	4	5
*m*1,2,3	0.1183	0.1773	0.3188	0.1773	0.2083

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
