# Peer review of "Fire Control System Operation Status Assessment Based on Information Fusion: Case Study"

_sensors, 2019, doi:10.3390/s19102222_

Reviewer 1 Report

Even if the authors changed several paragraphs and section compare to the previous submission, the quality of this paper seems poor.
The contribution has remained modest and some notations are not explained (which it makes difficult to understand the proposed approach). For instance:
1) Line 119: defined on B. Is it sure ?
2) Line 124: replace IND by Ind
3) Line 126: underline B is not defined
4) Line 141: replace V\par by V_\par
5) line 189: the symbol used for the emptyset is \emptyset (and not \Phi).
5) Line 191: replace \Phi by \Omega
6) Why the authors don't used an classical distance in the framework of belief function theory (Jousselme distance)?
7)...

Furthermore, this contribution contains some  grammatical and typographical errors. For instance, a lot of space are missed between words in the paper (lines 19, 37, 41, 50, 58, 92, 109, 131, 132, 145 173,...).

Author Response

Response to Reviewer 1 Comments was highlighted in yellow.

Point 1:Line 119: defined on B. Is it sure ?

Response 1:Line 118,19:defines the indistinguishable relation Ind(B) about B as:

Point 2:Line 124: replace IND by Ind

Response 2:Line 124 IND(A) replaced by Ind(A)

Point 3: Line 126: underline B is not defined

Response 3:Line 127: underline B is defined as the largest set of all subsets that must exist in the X set.

Point 4: Line 141: replace V\par by V_\par

Response 4: Line 142: is the value of the object  on the attribute C.

Point 5-1: line 189: the symbol used for the emptyset is \emptyset (and not \Phi).

Response 5-1: :lines 190,196,161,164: the symbol is revised to used for the emptyset is \emptyset.

Point 5-2:Line 191: replace \Phi by \Omega.

Response 5-2: Line 192 replace \Phi by \theta.

Point 6: Why the authors don't used an classical distance in the framework of belief function theory (Jousselme distance)?

Response 6:In evidence geometry space, it is relatively difficult to define a rigorous evidence distance measurement, which requires a lot of breakthrough background work and a lot of work. It is a simple and direct method to measure and estimate indirectly by using the method of indirect definition of similarity between evidences. In this paper, a method of calculating evidence similarity based on fuzzy membership degree is proposed.

Point 7:Furthermore, this contribution contains some  grammatical and typographical errors. For instance, a lot of space are missed between words in the paper (lines 19, 37, 41, 50, 58, 92, 109, 131, 132, 145 173,...).

Response 7:The space between words are supplemented,as follows:lines 19, 37, 41, 58, 92, 109, 132, 133, 146 ,174 , 210, 450, 459, 460, 468.But there isnt space is missed in line 50

Reviewer 2 Report

This paper proposed an information fusion by using evidence validity and evidence credibility to obtain the weight of evidence. Based on the proposed fusion method, the authors establish an intelligent evaluation model for the fire control system operation state.

Ps. I have reviewed the first version a couple of weeks ago and I confirm that this revised version carefully and well addresses my concerns. I am willing to recommend the acceptance of this work at its current status.

But I find some necessary editorial issues. For example, there are some repeated references, such as [50] and [51]. It should be 

Deng Y. Deng entropy. Chaos, Solitons & Fractals, 2016, 91: 549-553,

Finally, two latest works about conflict management could be mentioned as follows.

[1]  Zhang W, Deng Y. Combining conflicting evidence using the DEMATEL method. Soft Computing, 2018, DOI: 10.1007/s00500-018-3455-8 (https://doi.org/10.1007/s00500-018-3455-8).

[2] Wang Y, Zhang K, Deng Y. Base belief function: An efficient method of conflict management. Journal of Ambient Intelligence and Humanized Computing, 2018, DOI:10.1007/s12652-018-1099-2 (https://doi.org/10.1007/s12652-018-1099-2).

In short, I recommend to accept this work at this status. But suggest carefully revise some minor editiral problems at the proof stage.

Author Response

Response to Reviewer 1 Comments was highlighted in red.

Point 1: But I find some necessary editorial issues. For example, there are some repeated references, such as [50] and [51]. It should be Deng Y. Deng entropy. Chaos, Solitons & Fractals, 2016, 91: 549-553.

Response 1: The repeated references was revised in [52] .

Point 2: Finally, two latest works about conflict management could be mentioned as follows.

[1]  Zhang W, Deng Y. Combining conflicting evidence using the DEMATEL method. Soft Computing, 2018. DOI: 10.1007/s00500-018-3455-8(https://doi.org/10.1007/s00500-018-3455-8).

[2] Wang Y, Zhang K, Deng Y. Base belief function: An efficient method of conflict management. Journal of Ambient Intelligence and Humanized Computing, 2018, DOI:10.1007/s12652-018-1099-2 (https://doi.org/10.1007/s12652-018-1099-2).

Response 2: The first work “Combining conflicting evidence using the DEMATEL method” is mentioned in reference [41];

The second work “Base belief function: An efficient method of conflict management” was mentioned in reference [43].

Round  2

Reviewer 1 Report

Even if the article has improved, I am not wholly convinced. Moreover, there are still some errors and the bibliography must be update.

Author Response

Response to Reviewer 1 Comments was highlighted in green.

Point: Even if the article has improved,I am not wholly convinced,there are still some errors and the biblography must be updated.

Reponse 1: there are some repeated references, 1such as [4] and [42],the repeated reference was revised in [40];2[48]and[61],the repeated reference was revised in [45];3[34] and [55],the repeated reference was revised in [57] ;4[7]and[17] the repeated reference was revised in [15].

Reponse 2: Furthermore, this contribution contains a lot of space are missed between words in the biblography([1],[6],[7],[8],[9],[10],[12],[13],[14],[15],[16],[17],[18],[19],[21],[24],[25],[26],[27], [28],[30],[31],[48],[49],[50],[53],[55]; line109,110,111,115,124,125,139,162,166,181,182,217,385,427,428,448,473,482,497,506,507), the space between words are supplemented.

Reponse 3: line 132 :the term Ind (R)= Ind (R-{r} lose the“)”and it is revisied in line 132.

Reponse 4: line 198: equation (5),the punctuation of the parenthesis is revised.

Reponse 5: line 208,line 348,line 349: the location of the punctuation of . is revision.

Reponse 6: line  222 :“3.3.1 Causes of Conflict” and line 232“3.3.2. Classification of Conflict Problems”:  the format of the title is revised .

Reponse 7: line 331 : the repeated and the term mass function mwas deleted.   

Reponse 8:the biblography has been updated,such as the  revised references [1],[3],[5],[8],[10],[11],[12],[14],[16],[17],[22],[23],[24],[27],[28],[29],[31],[32],[33],[35],[36],[38],[41],[46],[47],[50],[54],[55],[57].

Reponse 9:the value of the outer border of the table 1-13 are revised.  

This manuscript is a resubmission of an earlier submission. The following is a list of the peer review reports and author responses from that submission.

Round  1

Reviewer 1 Report

This paper proposes yet another ad hoc combination way of merging belief functions, while the literature is already full of self-made ad hoc ones [2,3]. There are an infinite number of ways for building belief functions from several belief functions. What is important is the interest of the combination operator, its justification, its fundamentals, its properties.

Unfortunately, in this paper, the "new method" is not all justified and consists in a succession of ad hoc operations. What are the expected properties for this new method? It is not clear. Consider the example given on page 8 and Table 3. The authors do not explain mathematically what are the axioms, what are the properties the result should fulfill.

The article is also very poorly written (Numerous English mistakes, very difficult to read). Need scientific backgrounds too.

Reference [1] should give solid backgrounds on belief functions.

References [2,3,4,5] should detail what is expected to introduce combination operator.

[1] Glenn Shafer, A Mathematical Theory of Evidence, Princeton University Press, 1976.

[2] Philippe Smets, Analyzing the combination of conflicting belief functions, Information Fusion, Volume 8, Issue 4, 2007, Pages 387-412.

[3] Didier Dubois, Weiru Liu, Jianbing Ma, Henri Prade, The basic principles of uncertain information fusion. An organised review of merging rules in different representation frameworks, Information Fusion, Volume 32, Part A, 2016, Pages 12-39,

[4] Sébastien Destercke, Thomas Burger, Toward an axiomatic definition of conflict between belief functions, IEEE Transaction Systems Man Cybernetics part B, 2012, 43 (2), pp.585-596

[5] Frederic Pichon, Sébastien Destercke, Thomas Burger, A Consistency-Specificity Trade-Off to Select Source Behavior in Information Fusion, IEEE transactions on systems, man, and cybernetics, Institute of Electrical and Electronics Engineers (IEEE), 2015, 45 (4), pp.598-609.

Reviewer 2 Report

  This manuscript proposed a new information fusion method to identify the state type of the system in operation based on rough sets theory and improved D-S evidence theory. Rough sets theory is used to reduce attribute and obtain BPAs. Authors improve D-S evidence and use it to fuse the obtained BPAs. The improved method of conflict evidence combination seems effective. But it is written in very bad English and language correction is needed, the organization of this paper also need improvement. Other comments:

1.       In Abstract, “d-s evidence” should be “D-S evidence”.

2.       In Section 1, information about rough set theory should be introduced more.

3.       In Section 1.1, line 99, “A=C∩D” should be "A=C∪D”.

4.       In Section 1.1, line 100, what is the information systemⅠ?

5.       The format of the title is confusing. Some titles don't have uppercase at all, such as “attribute reduction”, “acquisition method of basic probability value”, and “weight distribution of conflict evidence”. The first letter of the first words of some titles are capitalized, such as “Total conflict”, “Classification of conflict problems”. The first letter of each word in some titles is capitalized, such as “Information System and Decision Table”, “Establish a Diagnosis model”.

6.       Section 1.2 just simply introduced the attribute reduction, the role of rough sets in attribute reduction is not explained.

7.       In Section 2.1, line 125, “” is lost. And line 126, “Then m can be called the basic credibility…” should be in a new line.

8.       In Section 2.1, equation (2) is not clear. “Where A and B are the basic credibility functions…” But there are not “A” and “B” in equation (2). And what does m(C) means?

9.       In Section 3.1, equation (3) is wrong, missing minus sign. And Shannon entropy cannot measure the uncertainty of a BPA well. It is recommended to use Deng entropy.

10.   In Section 3.1, equation (6), what does crd(m_i) means?

11.   In Section 3.2, line 237, in “Let there be identification frame O”, “O” should be “Θ”.

12.   In Section 3.2, line 243, in “Where, V represents intersection (smaller value) and U represents union set (larger value).” “V” should be “∧”, “U” should be “”. There are many errors in equations of this manuscript, the author should examine carefully.

13.   In Table 3, “m (B)” should be “m(B)”.

14.   Figure 1 should show the roles of rough set theory and D-S evidence theory rather than two separate parts.

15.   The whole manuscript is missing citations, only the Introdution have some citations.

16.   The reference could be updated and some recent progress on conflicting evidence management and data fusion should be mentioned as follows.

[1] X. Su, L. Li, F. Shi and H. Qian, "Research on the Fusion of Dependent Evidence Based on Mutual Information," in IEEE Access, (2018). https://ieeexplore.ieee.org/abstract/document/8542666

[2] LI, Zhen; CHEN, Luyuan. A novel evidential FMEA method by integrating fuzzy belief structure and grey relational projection method. Engineering Applications of Artificial Intelligence, 77 (2019) 136-147.

[3] Chen, L., & Deng, X. A Modified Method for Evaluating Sustainable Transport Solutions Based on AHP and Dempster–Shafer Evidence Theory. Applied Sciences, (4) (2018) 563.

[4] Y. Li, Y. Deng, Generalized ordered propositions fusion based on belief entropy, International Journal of Computers Communications & Control 13 (5) (2018) 792–807.

[5] Zhang H, Deng Y. Engine fault diagnosis based on sensor data fusion considering information quality and evidence theory. Advances in Mechanical Engineering. 10 (10) (2018): doi10.1177/1687814018809184.

Based on above evaluation, I recommend to MAJOR REVISE of this paper.

Reviewer 3 Report

This paper deals with the problem of diagnosis methods. The authors propose to use the rough set theory combiend with the evidence theory to identify the state type of the system.
The rough set theory is introduced to reduce the number of attibute and to obtain a bbas. A new combination rule is proposed to combine the bbas in the framework of belief function.
An example of the proposed approach on the intelligent evaluation model of the power supply module of a fire control computer is presented.

The contribution is relatively poor and proposals are not theoretically justified. Many articles deal with the conflict redistribution and few of these papers are cited.

Some classical references of belief functions are omitted. For example in section 2.4, the authors could add:
P. Smets: Analyzing the combination of conflicting belief functions. Information Fusion, 8(4):387–412, 2007.
E. Lefevre, O. Colot, P. Vannoorenberghe: Belief function combination and conflict management. Information Fusion 3(2): 149-162 (2002)
In section 2.5:
L. Zadeh: A simple view of the Dempster–Shafer theory of evidence and its implication for the rule of combination, AI Magazine 7 (1986) 85–90.
In Section 3.1:
N. R. Pal, J. C. Bezdek, R. Hemasinha: Uncertainty measures for evidential reasoning I: A review International Journal of Approximate Reasoning, Volume 7, Issues 3–4, October–November 1992, Pages 165-183
N. R. Pal, J. C. Bezdek, R. Hemasinha: Uncertainty measures for evidential reasoning II: A new measure of total uncertainty International Journal of Approximate Reasoning, Volume 8, Issue 1, January 1993, Pages 1-16
Eq. 1 is not enough explain to be understand.
In eq. 4, why only the overlined term is squared?
Instead of eq. 8, why the authors don't used an classical distance in the framework of belief function theory (Jousselem distance)?